# Multi-task machine learning improves multi-seasonal prediction of the Indian Ocean Dipole

Fenghua Ling[1], Jing-Jia Luo [1] ✉, Yue Li[1], Tao Tang[1], Lei Bai[2], Wanli Ouyang[2,3] & Toshio Yamagata [1,4]

As one of the most predominant interannual variabilities, the Indian Ocean Dipole (IOD) exerts great socio-economic impacts globally, especially on Asia, Africa, and Australia. While enormous efforts have been made since its discovery to improve both climate models and statistical methods for better prediction, current skills in IOD predictions are mostly limited up to three months ahead. Here, we challenge this long-standing problem using a multi-task deep learning model that we name MTL-NET. Hindcasts of the IOD events during the past four decades indicate that the MTL-NET can predict the IOD well up to 7-month ahead, outperforming most of world-class dynamical models used for comparison in this study. Moreover, the MTL-NET can help assess the importance of different predictors and correctly capture the non-linear relationships between the IOD and predictors. Given its merits, the MTL-NET is demonstrated to be an efficient model for improved IOD prediction.

The Indian Ocean Dipole (IOD) is a dominant interannual climate variability in the tropical Indian Ocean involving strong air-sea coupling. It starts in late boreal spring with a gradually shoaling thermocline as a response to the anomalous equatorial zonal winds in the Indian Ocean. The west-east gradient of sea surface temperature (SST) anomalies resulting from the shoaling thermocline further reinforces the zonal wind anomalies via Bjerknes feedback, leading to a culminated IOD event in autumn. With the reversal of monsoonal wind, IOD decays rapidly in early winter. Conventionally, the Dipole Mode Index (DMI) is used to identify IOD events. The DMI is defined by the west-east gradient of SST anomalies between the equatorial western Indian Ocean and the southeastern Indian Ocean[1–3]. Through a variety of atmospheric and oceanic passages, the IOD can profoundly influence weather and climate in many regions, especially the countries in the rim of the Indian Ocean, such as Australia, India, East Africa, and East Asia[3–6]. Due to its wide-spread socio-economic impacts, active international efforts have been made in past decades since its discovery to improve the IOD prediction by developing/updating atmosphere-ocean coupled model forecast systems and/or statistical methods[7–12].

However, skilful prediction of the IOD still remains a long-standing challenge[13]. Compared to the prediction of El Niño/Southern Oscillation (ENSO) events, which now can be extended up to 18 months ahead[14,15], the prediction of the IOD events (except for the super IOD event in 2019) is still limited to three months ahead in many current state-of-the-art climate model forecast systems[13,16,17]. The major difficulties in predicting IOD are rooted in a variety of unique characteristics of the tropical Indian Ocean. The relatively small Indian Ocean basin is bounded by the Eurasian Continent to the north and occupied with warm water over which stochastic convections are frequent. The Indian Ocean is strongly impacted by active intra-seasonal oscillations that are mostly unpredictable beyond a few weeks[13]. In contrast to the ENSO in Pacific Ocean, the air-sea coupling in the Indian Ocean is weak with the prevailing westerlies driving a deeper thermocline in the east than in the west. In addition, multi-scale nonlinear interactions are active in the eastern Indian Ocean. All these processes are systematically underestimated in climate models[18].

Owing to the complicated mechanisms and nonlinear processes intrinsic to the IOD, many traditional statistical models are also

[1]Institute for Climate and Application Research (ICAR)/CIC-FEMD/KLME/ILCEC, Nanjing University of Information Science and Technology, Nanjing, China. [2]Shanghai AI Laboratory, Shanghai, China. [3]School of Electrical and Information Engineering, The University of Sydney, Sydney, NSW, Australia. [4]Application Laboratory, Japan Agency for Marine-Earth Science and Technology, Yokohama, Japan. ✉e-mail: jjluo@nuist.edu.cn

suffered for the limited prediction skills[7,8]. With the advent of the big data era, deep learning methods have displayed great potential in predicting weather and climate by detecting intricate structures hidden in large datasets. For instance, the deep learning methods have been widely applied in recent years to predict ENSO, synoptic precipitation and many other phenomena, demonstrating comparable or even advantageous ability relative to dynamical model prediction systems[11,12,15,19,20]. Here, we develop a model based on machine learning technique to predict the IOD events in their peak season (i.e., September-October-November; SON) as well as their onset and development seasons (i.e., March-April-May and June-July-August). We believe that the robust analyses based on the climate mechanisms may further contribute to an increased credibility of the machine learning model.

## Results

### Improved prediction of SON DMI with MTL-NET

It has been widely recognized that ENSO and IOD are strongly influenced by each other owing to the interactions among the tropical oceans[21,22], this inter-basin coupling needs to be properly considered in the prediction model. Distinct from previous efforts using the machine learning method for a single task[11,12,15,19,20], we utilise a multi-task framework, namely MTL-NET, in order to capture the important interactions between the Indian Ocean and the Pacific Ocean. In the MTL-NET, along with the primary task to predict the DMI, we also set three secondary tasks to predict, namely EIOD, WIOD and Nino3.4 indices. The EIOD and WIOD indices are the SST anomalies averaged over the eastern pole of the Indian Ocean (50°–70°E,10°S-10°N) and the western pole of the Indian Ocean (90°–110°E,10°S-0°), respectively. The ENSO-related Nino3.4 index is the average of SST anomalies over the region 5°N-5°S and 170°–120°W.

Since the IOD is driven by intrinsic ocean-atmosphere coupled processes, SST, heat content (i.e., vertically-averaged ocean temperature in upper 300 m), surface zonal (Us) and meridional wind (Vs) anomalies covering the region of 0°–360°E, 55°S–60°N during three

consecutive months prior to the forecast start month are selected as the predictors to feed the MTL-NET (Fig. 1, see "Methods"). Adding the surface wind predictor helps improve the IOD prediction skill particularly at lead times beyond 7 months (Supplementary Fig. 1). To consider the high persistence of the tropical climate signals, we have inserted a Long Short-Term Memory (LSTM) block to capture their temporal relations (Fig. 1). In fact, the added LSTM block assists MTL-NET in outperforming other popularly used Convolutional Neural Networks (CNN) models[12]. The MTL-NET displays the highest performance in predicting the IOD among various machine learning models (Supplementary Fig. 1, "The MTL function and advantages").

Figure 2 further highlights the advantage of the MTL-NET for the DMI predictions compared to fourteen world-class state-of-the-art dynamical model prediction systems including seven operational models. Both the correlation skills and root mean square errors (RMSEs) in predicting the DMI by the MTL-NET are persistently superior to most of the fourteen dynamical counterparts at lead times of up to 7 months and even beyond. The prediction skills of the DMI based on 13 out of 14 dynamical models are lower than the MTL-NET at 1-month lead. With the increase of lead time, the skills of most dynamical models decrease rapidly. Although some dynamical models have comparable skills at 1–4 months lead, their prediction skills become lower than that of the MTL-NET afterwards. It is worth noting that the MTL-NET can predict SON DMI at lead time beyond 12 months with a correlation skill close to 0.4 up to 15-month lead. In addition, the MTL-NET also outperforms most of the fourteen dynamical models in predicting the IOD in boreal spring and summer (Supplementary Fig. 2), indicating its improved skills in predicting the onset and development of the IOD events. The predicted DMI timeseries at lead time of 1, 3, 6 and 12 months based on MTL-NET demonstrates that the MTL-NET can predict many of the IOD events during 1983–2019 (Fig. 2c, Supplementary Fig. 2c and d).

As an advantage of the multi-task framework, if setting another task (e.g., the EIOD and WIOD indices) as the primary task, it can

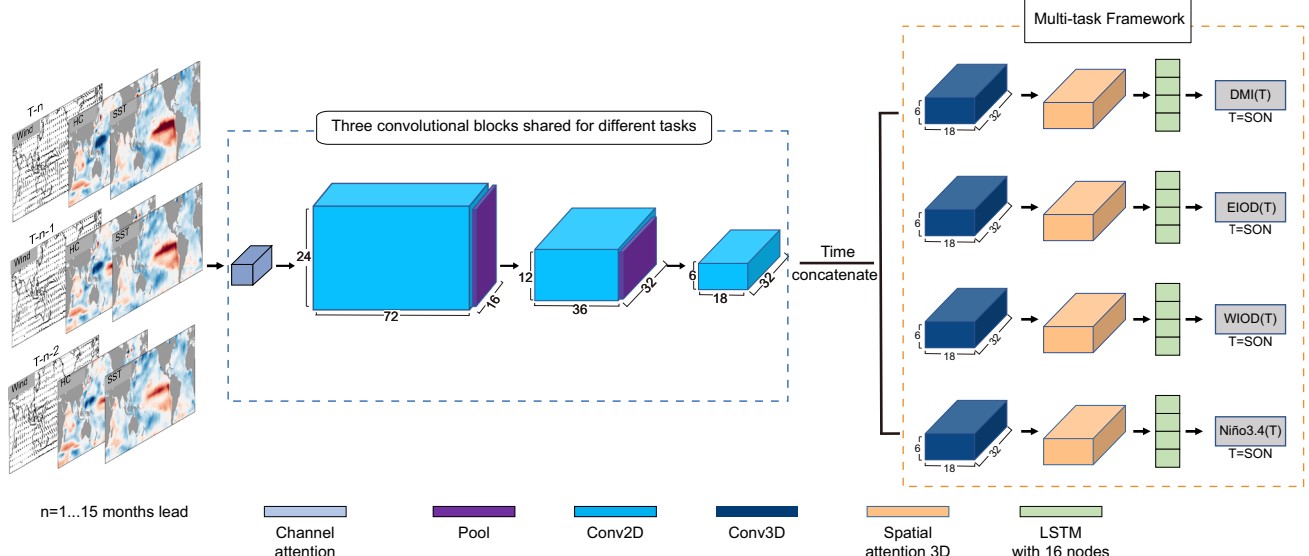

**Fig. 1 | The structure of the multi-task learning model (MTL-NET).** The MTL-NET contains three input layers (i.e., the predictors in each month) and each input will use the convolutional block to extract spatial features. The convolutional block consists of one channel attention layer, three convolutional layers and two maximum pooling layers. Then, the spatial features in each month are concatenated according to the time dimension and fed into the different tasks. Each task block contains a 3d convolutional layer, a spatial attention layer, a long short-term memory (LSTM) layer, and an output layer (i.e., dense layer, the predictand). The variables of the input layer include sea surface temperature (SST; °C), upper 300 m heat content (HC; °C), surface wind (Us and Vs; m s⁻¹) anomaly from T-n-2 months to T-n months over the globe (0°–360°E and 55°S–60°N). T denotes the target season (i.e., Sep-Oct-Nov, SON) and n denotes the lead time from 1 to 15 months. The lead time is defined as the number of months between the latest available observations and October (i.e., the middle month of SON). The Dipole Mode Index (DMI), the SST index of eastern pole of the Indian Ocean (EIOD), the SST index of western pole of the Indian Ocean (WIOD), and Nino3.4 SST index (Nino3.4) in SON are selected as the predictands.

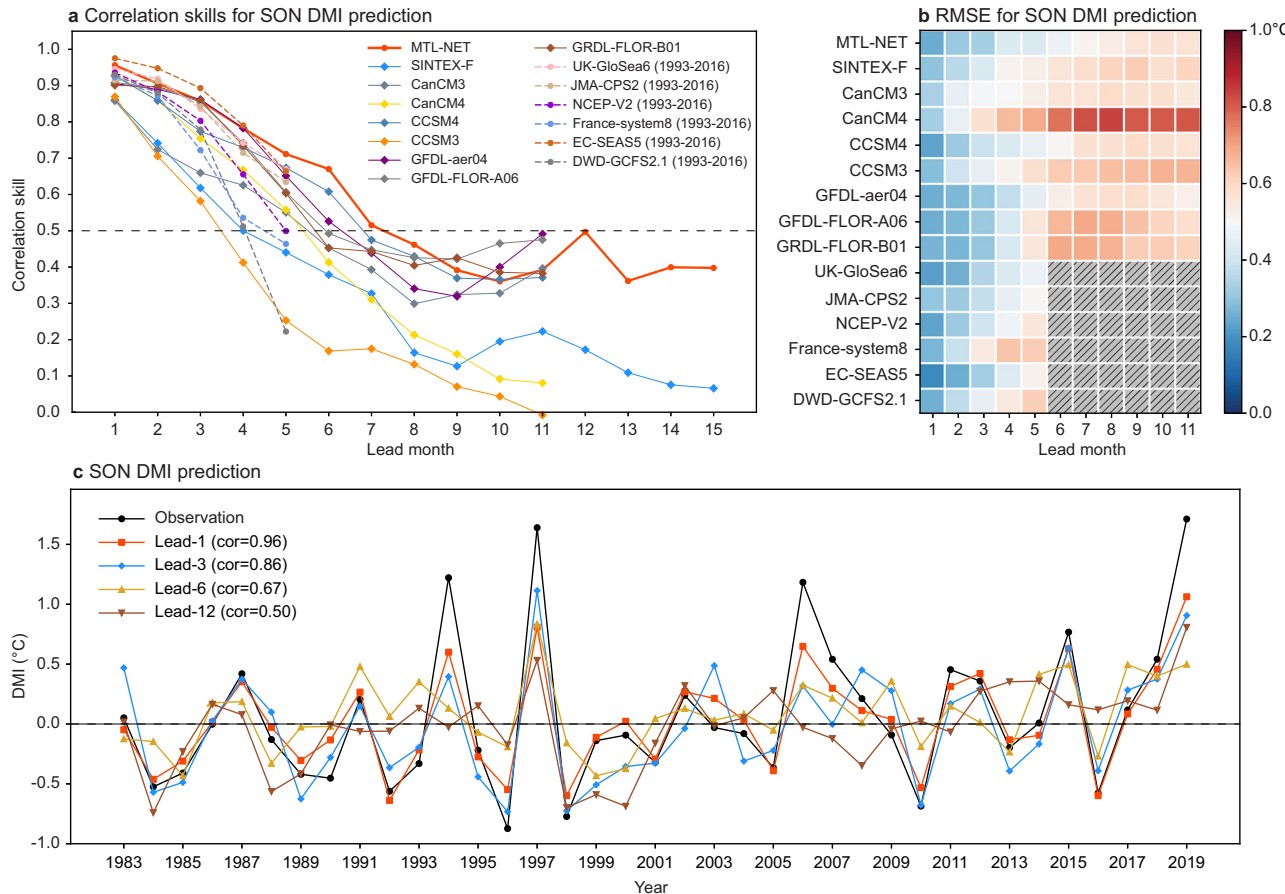

**Fig. 2 | Correlation skills of the Indian Ocean Dipole prediction. a** Predictive skill of the Dipole Mode Index (DMI) in Sep-Oct-Nov as a function of the lead month based on the multi-task learning model (MTL-NET; red line), SINTEX-F dynamical forecast system (blue line), six operational forecast models which hindcast periods are limited to the period after 1993 (dashed lines) and seven dynamical forecast systems of the North American Multi-Model Ensemble (NMME) project (the other coloured lines). The prediction skill is validated for the period of 1983–2019. Black dashed line denotes the skill of 0.5. **b** Root mean square error (RMSE) of the SON DMI prediction based on the MTL-NET and each dynamical model forecast system. The diagonal line represents the model does not have prediction at this lead time. **c** The SON DMI based on the observations (black line) and 10-member ensemble mean predictions of MTL-NET at lead time of 1, 3, 6 and 12 months (red, blue, yellow and brown lines), respectively.

simultaneously improve the predictions that are also superior to most of the fourteen dynamical models' performance (Supplementary Fig. 3). Furthermore, the predictions of the MTL-NET appear to be rather robust against different test datasets (Supplementary Fig. 4). Therefore, we conclude that the MTL-NET provides a reliable forecast of the IOD events up to 7 months ahead and even beyond, which has yet not been achieved in most of the fourteen dynamical forecast systems.

**Precursors revealed by MTL-NET for strong IOD events**
Furthermore, the channel attention mechanism (see "The attention blocks"), by estimating the time-varying importance of every predictor at each lead time, helps understand the reasons why such a reliable IOD prediction can be extended to long lead times (Supplementary Fig. 5). Among the four predictors, the importance of the SST anomaly is remarkable at 1-month lead time because the SST dipole in the Indian Ocean has already formed in July-August-September (JAS) that is about 1-month prior to the IOD peak season SON. With the increase of lead time, the zonal and meridional winds become more important up to 3-month lead compared to the other predictors. This is consistent with previous studies that suggested the strong impacts of the summer monsoon on the development of IOD[1,3]. As expected, the importance of the heat content anomalies of the upper ocean to the IOD prediction becomes more remarkable with the increase of lead time due to strong oceanic memory[9]. This is different from the varying importance of the other predictors among different lead times.

It is worth noting that the predictability of IOD at each lead time may come from the precursors in different regions. It is also interesting and important to explore the possible precursors that reinforce the intensity of an IOD event, especially during the development of an IOD event. Therefore, the heatmap derived from the spatial attention mechanism (see "The attention blocks") is utilised to find out the key regions at 3-month lead, which generally represents the current prediction level in many dynamical forecast models. A positive (negative) IOD event is defined as the DMI being greater (lower) than one positive (negative) standard deviation during SON. Accordingly, five strong positive IOD events (pIOD; i.e., 1994, 1997, 2006, 2015, and 2019) and five strong negative IOD events (nIOD; i.e., 1992, 1996, 1998, 2010, and 2016) since 1983 are selected. They are composited separately to demonstrate the distinct mechanisms in the pIOD and nIOD predictions.

Here, a larger value in the heatmap denotes a greater contribution of the regional predictors to the IOD prediction (Figs. 3a and 3b). The heatmap of 3-month lead prediction (i.e., initiated from May-June-July (MJJ)) suggests that precursors in four regions including the tropical western Pacific, Australia, the North and Southeast Pacific are crucial to the successful pIOD predictions (Fig. 3a). The obvious westerly anomaly in the equatorial western Pacific may trigger a downwelling Kelvin wave and transport warm water eastward. In due course, the

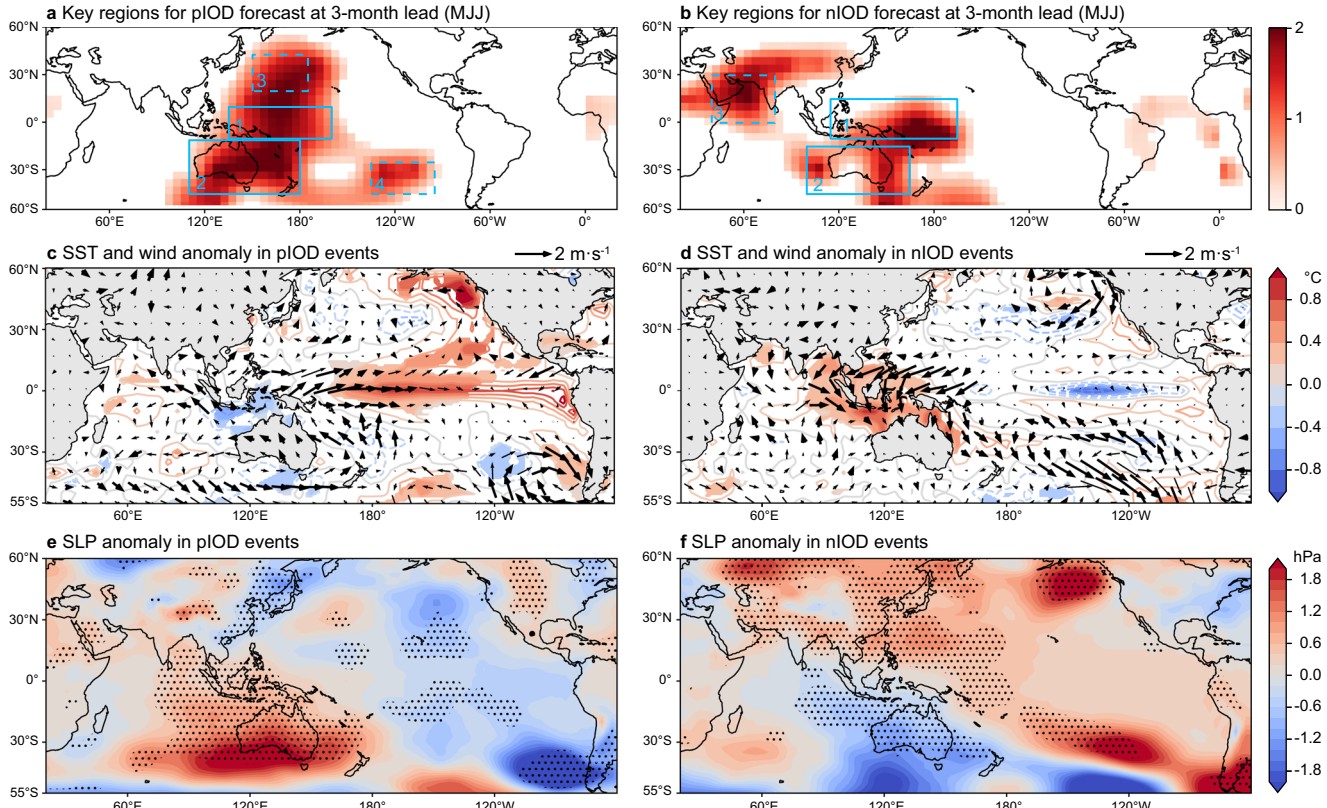

**Fig. 3 | Physical interpretation of the Indian Ocean Dipole (IOD) predictions at 3-month lead. a** Spatial attention map of the five strong positive IOD events (pIOD) during 1983–2019 based on the 3-month lead predictions, i.e., initiated from May-June-July (MJJ). Only the values statistically significant at 10% level are displayed. The heatmap values are split into different key regions (i.e., blue boxes) according to known physics. The composite maps of (**c**) sea surface temperature (SST; shading and contour) and surface wind (Us, Vs; vector) and (**e**) sea-level pressure (SLP) anomalies in MJJ during the five strong pIOD years. (**b**, **d**, **f**) As in (**a**, **c**, **e**), but for the results of the five strong negative IOD events (nIOD). The vectors in bold, stippling and shading in (**c**–**f**) denote the areas where the anomalies are significant at 10% level based on the Student's *t* test.

warm SST anomaly in the central-eastern Pacific Ocean will strengthen, while the Walker circulation will weaken due to the Bjerknes feedback. The eastward movement of the deep convection results in the abnormal divergence in the western tropical Pacific (region 1 in Fig. 3a) and the weakening of the climatological westerly wind in the Indian Ocean. Therefore, it benefits the formation and growth of the pIOD events[9,17,21–24] (Fig. 3c and Supplementary Fig. 6a).

In addition, the strengthening of the Australian High (region 2 in Fig. 3a) reinforces the cross-equatorial flow over the Maritime Continent (Fig. 3e), which intensifies the coastal upwelling along the Sumatra during boreal summer, and subsequently contributes to the cold SST anomaly over the southeastern tropical Indian Ocean[24] (Supplementary Fig. 6b). Note that the far-fetched significant sea-level pressure (SLP) gradient between the northwestern Pacific (region 3 in Fig. 3a) and the equatorial western Pacific, which is nonlinearly correlated with the pIOD events (Supplementary Fig. 6c), might mostly represent a response to the equatorial Pacific warming learnt by the MTL-NET, rather than a contributor to the pIOD predictability. Similarly, the signal in the Southeast Pacific (region 4 in Fig. 3a) resembles the Pacific South American pattern[25], which is also a response to the SST warming in the equatorial central-eastern Pacific (Fig. 3c and e).

In the nIOD predictions at 3-month lead, the key regions of the precursors locate in the tropical western Pacific, Australia, and the Arabian Sea (Fig. 3b). The significant northeasterly wind anomaly in the western equatorial Pacific contributes to the nIOD events in two ways. In one way, the strengthened northeasterly wind may intensify the Indonesian throughflow, and transport more warm pool water from the tropical western Pacific into the eastern Indian Ocean[26]. In the

other way, the enhanced surface convergence over the warm pool may strengthen the climatological westerly wind in the Indian Ocean, and thus results in the nIOD events. Moreover, there is a negative SLP anomaly in Australia that weakens the northward cross-equatorial wind due to the decreased interhemispheric SLP gradient (Fig. 3d and f), therefore favouring the occurrence of nIOD events[22,24] (Supplementary Fig. 7a, b). The warm SST and northerly anomalies in the Arabian Sea appear to be driven by preceding El Niño event through the Indian Ocean Capacitor mechanism[27], just as was shown in the 1998 nIOD event (Supplementary Fig. 7c, d). The warm SST anomaly in the tropical Indian Ocean forces a basin-wide cyclonic circulation, which is reminiscent of the classical Matsuno-Gill type of atmospheric response to a tropical heating. The associated westerly anomalies along the equatorial Indian Ocean help promote the development of nIOD events.

It is interesting to find a rebound of the correlation skill in the MTL-NET and a few dynamical models at 10–12 months lead predictions (Fig. 2a), and this may indicate some multi-seasonal predictability of the IOD. To understand this, we have calculated a heatmap of the MTL-NET to identify the key regions of the precursors for the 12-month lead predictions (Fig. 4). Precursors in two key regions are found to be important to the multi-seasonal predictions of the pIOD events (Fig. 4a). In the central North Pacific (region 1 in Fig. 4a), an apparent negative SLP (i.e., cyclone) anomaly appears in August-December of the previous year (Supplementary Fig. 8). The southwesterly wind over the southern side of the cyclone weakens the northeasterly trade wind, reducing surface evaporation, and hence increasing the SST in the western coast of North America (Supplementary Fig. 9). Meanwhile,

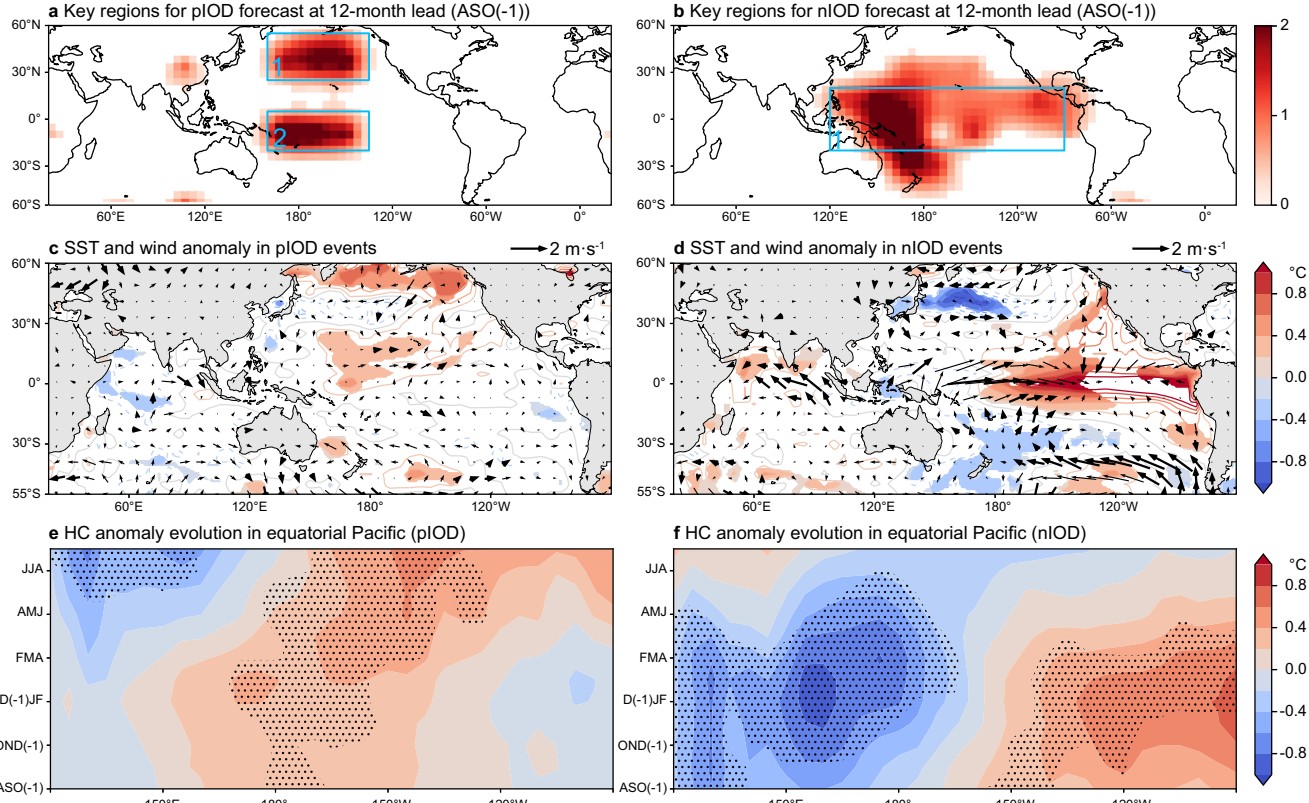

**Fig. 4 | The physical interpretation of the Indian Ocean Dipole (IOD) predictions at 12-month lead. a** Spatial attention map of the five strong positive IOD events (pIOD) during 1983–2019 based on the 12-month lead predictions, i.e., initiated from August-September-October of the previous year (ASO (−1)). Only the values statistically significant at 10% level are displayed. The heatmap values are split into different key regions (i.e., blue boxes) according to known physics. (c) Composite map of the observed sea surface temperature (SST; shading and contour) and surface wind (Us, Vs; vector) anomalies during ASO (−1) of the five strong pIOD years. **b**, **d** As in (**a**, **c**), but for the results of the five strong negative IOD events (nIOD). **e** The evolution of the upper 300 m heat content (HC) anomaly along the equatorial Pacific (5°S-5°N) for the five strong pIOD events. **f** As in (**e**), but for the results (10°S-10°N) of the five strong nIOD events. The vectors in bold, stippling and shading in (**c**–**f**) denote the areas where the anomalies are significant at 10% level according to the Student's *t* test.

the weakening of the northeasterly trade wind reduces the convergence along the intertropical convergence zone at around 10˚N, and the decreased convective precipitation and cloud cover lead to more downward surface solar radiation to increase the SST. Therefore, there is a southwest-northeast oriented warm SST anomaly extending from the equator to the western coast of North America[28,29] (Fig. 4c and Supplementary Fig. 9).

The seasonal foot-print mechanism triggered by the strong air-sea interactions allows this spatial pattern to persist from boreal winter to the following spring and summer[28–31]. It also promotes the occurrence of a central-Pacific type of El Niño during August-September-October of the previous year (ASO (−1)) to January-February-March (JFM) that subsequently moves eastward in following seasons (Fig. 4c, e). The eastward propagation of warm upper ocean HC anomalies along the equatorial Pacific brings warm water up to the surface[28–32] (Fig. 4e and Supplementary Fig. 9) and weakens the Walker circulation. The latter leads to the slackening of the climatological westerly wind in the Indian Ocean, and thus favours the occurrence of pIOD events. We conclude that the MTL-NET can capture the initial features of the seasonal foot-print mechanism that provides the robust 12-month lead precursors for the occurrence of strong pIOD events, a finding that has not been disclosed in previous studies.

Distinctively, the key region of the 12-month lead precursors for the nIOD events mainly confines within the tropical Pacific (Fig. 4b). The heat content evolution along 10˚S-10˚N suggests a clear phase transition from an El Niño to a La Niña during ASO (−1) to JAS (Fig. 4d, f and Supplementary Fig. 10). Associated with the development of a La Niña, both the Indonesian throughflow and the Walker circulation become stronger than normal, transporting more warm water from the western tropical Pacific to the Indian Ocean, and thus increasing the SST in the eastern Indian Ocean[22,27,33]. This result indicates that the preceding El Niño signal and the following regular phase transition to a La Niña provide an important precursor for the 12-month lead forecast of the nIOD events.

## Discussion

Recently, the deep learning method has been widely utilised for weather-climate predictions although its interpretability is required for further improvement. Our study suggests that deep learning methods can extend reliable IOD predictions out to 7 months ahead, and the combination of the deep learning method with the geophysical big data can help deepen our understanding of complex climate variabilities in the Earth system. Traditional analyses largely rely on linear regression to explore the relationship between the predictors and predictands and the nonlinear relationships are often neglected. The deep learning method can be an effective complementary as the deep learning model is built on a series of nonlinear calculations. Based on the interpretable analyses of our model results, we can obtain the distinctive mechanisms responsible for positive and negative IOD events. This helps deepen the understanding of the nonlinear mechanisms of IOD, although they need to be fully tested by traditional dynamical model experiments and other approaches. Indeed, a recent finding[29] supports our deep learning model results on the importance of the North Pacific precursors at 12-month lead. In future

work, we may focus on addressing the interpretability of how to separate the different precursors at different lead times. Perhaps, extracting high-dimensional features of multi-modal dataset will help address this issue.

# Methods

## Datasets

Centennial historical simulations from Coupled Model Inter-comparison Project phase 5 and 6 (CMIP5 and 6)[34,35] and reconstructed historical observation data[36,37] were used to train the MTL-NET (Supplementary Table 1 and 2). They are used to meet the requirement of big data for training the artificial intelligence model. To validate the performance of the model by comparing with the observed values, monthly mean SST and upper 300 m heat content data were collected from the Global Ocean Data Assimilation System (GODAS) reanalysis during 1983–2019[38], while the horizontal wind vector data were obtained from the NCEP-DOE Reanalysis 2[39]. We first built a Convolutional Neural Networks (CNN) prediction model using only CMIP5 simulations and using both CMIP5 and CMIP6 outputs, respectively. When building the MTL-NET, we tested the model's performance by dropping off surface winds, and both surface winds and LSTM block, respectively. Finally, we built the MTL-NET by including both the LSTM and the four ocean-atmosphere predictors. The result reveals that more training data can produce better prediction skill and adding surface winds as predictors helps improve the skill, especially in predicting the strong IOD events (Supplementary Fig. 1). The MTL-NET provides the best prediction skill among all the built deep-learning models.

## NMME and SINTEX-F hindcasts

The North American Multi-Model Ensemble (NMME) is an experimental project, which was established in response to the U.S. National Academies' recommendation to support regional climate forecasting and decision-making over intra-seasonal to interannual timescales. The project has been contributing model predictions from their hindcasts (dating back to the early 1980s) and real-time forecasts since August 2011. Each model consists of 6–28 ensemble members, and the forecasts are provided at lead times from 1 month to 11 months[40] (https://iridl.ldeo.columbia.edu/SOURCES/.Models/.NMME/).

The SINTEX-F prediction system is built based on a fully coupled global ocean–atmosphere circulation model developed under the EU-Japan collaborative framework. This system has displayed high performance in predicting the tropical climate signals[41]. In particular, several ENSO events can be predicted at lead times of up to 2 years by this system[14]. The real-time predictions have been updated every month and made publicly available since 2006 (see http://www.jamstec.go.jp/aplinfo/sintexf/e/seasonal/outlook.html and https://icar.nuist.edu.cn/en/111/list.html).

## Operational forecast models' hindcasts

The Copernicus Climate Change Service provides a multi-system seasonal forecast service, where data produced by state-of-the-art seasonal forecast systems developed, implemented and operated at forecast centres in several countries is collected. The data includes the retrospective forecasts (hindcasts) during the period 1993–2016, and the forecasts are provided at lead times from 1 month to 5 months (https://cds.climate.copernicus.eu/cdsapp#!/dataset/seasonal-monthly-single-levels?tab=overview).

## The MTL-NET

We built the MTL-NET to predict the IOD. A key challenge is how to capture the spatiotemporal dependencies simultaneously with one single forecast model and improve prediction skills of the IOD at seasonal-to-multi-seasonal lead times. To build the MTL-NET at $n$ months lead ($n$=1, 2, 3, …, 15) for the target season (i.e., SON), we used

both the ocean and atmosphere predictors over the globe with a 5°×5° horizontal resolution during three consecutive months prior to each forecast start month. To better capture the spatial features, we used a convolution module[42] and appropriately add maximum pooling layers to extract the most important regional features. To capture the temporal features, we added a LSTM[43] module after extracting the spatial feature. It is worth noting that ENSO often plays an important role in the evolution of the IOD[21,22]. The inter-basin interactions between the Pacific and Indian Oceans can strongly impact the climate variations in the Indian Ocean. Therefore, we use the multi-task learning framework (MTL) contains both the IOD indices and Nino3.4 SST index in the prediction model. Such a framework allows the model to share the extracted spatiotemporal features from both ENSO and the IOD. Thus, the important inter-basin coupling between the Pacific and the Indian Ocean is included in the MTL. In addition, to better understand the model results and mechanisms underpinning the seasonal-to-multi-seasonal predictability of the IOD, we added the attention blocks. To avoid unstable predictions of MTL-NET, the results are produced based on the integrated learning from 10 ensemble members of the MLT-NET model for each lead time. The construction and workflow of the MTL-NET is schematically displayed in Figure 1.

## The MTL function and advantages

When using a supervised learning to build a prediction model, a usual way is to train one model for each single task separately[11,12]. However, the Earth is a holistic system and this may miss the important interactions between the different elements in the system, especially in the tropical oceans where strong multi-scale interactions operate. The MTL is capable of sharing parameters between multiple tasks to a certain degree, and hence can improve the original single prediction task[44]. Indeed, Supplementary Fig. 1 displays that the MTL-NET performs better than the single-task model in predicting the IOD during SON. In addition, the prediction skill of the MTL-NET reaches 0.73 at 7 months lead, which is higher than that of a single-task model[12] (i.e., 0.69) if following the same evaluation way.

The main feature of the MTL is that it can deal with multiple predictands (i.e., tasks) and different tasks can share loss functions according to the task weights[45]. Here, we mainly adopted mean square error (MSE) for the loss function. In the multi-task model, the loss function is written as follows.

$$LOSS = \sum_{i=1}^{n} a_i \cdot L_i \qquad (1)$$

where $a_i$ and $L_i$ is the weight and loss of the $i$-th task, $n$ is the number of the tasks. The prediction performance of the MTL-NET is affected by the weight values. When building the MTL-NET for the SON DMI prediction, we use four tasks including the DMI, EIOD, WIOD, and Nino3.4 SST indices. Note that the primary task (i.e., DMI) is harder to learn with a low predictability, the auxiliary task (Nino3.4) is easier to learn with a high predictability, and the IOD and ENSO are highly related due to the strong Indo-Pacific inter-basin interactions. Thus, the MTL can obtain the better predictions of DMI with helps of the auxiliary task[46]. And we have conducted two additional experiments by swapping the primary and secondary predictands (Supplementary Fig. 11). The result demonstrates that the IOD prediction skills as the primary task are generally better than those as the secondary task. In addition, the IOD prediction skills calculated using the EIOD and WIOD indices are lower than those of the IOD as the primary task. This is also true whether the EIOD or WIOD is taken as the primary task (Supplementary Fig. 1).

This function can help the model pay more attention to a certain task and extract more features about this task by optimizing the value of the weights to obtain better forecast skills. For example, with

the increase of lead time, the role of the Pacific Ocean may become more and more important, and hence we can obtain more features about the Pacific Ocean by increasing the weights of Nino3.4 task based on the premise that the primary task is set for the prediction of the DMI.

We also designed an experiment with the same parameters but removing the influence of ENSO. The skill drops at 5-month lead and beyond, becoming much lower than the MTL-NET (Supplementary Fig. 1a). As for the importance of the predictors revealed by the channel attention and the heat map, the results are generally similar (figures not shown). However, the results based on the MTL-NET with ENSO task provides clearer and more area-focused information on the importance of the predictors at different lead times.

In addition, we set the weight of the DMI task to 5 and the weights of the other tasks to 1 at 1-month lead, and allow the weight of Nino3.4 task gradually increase with lead time (Supplementary Fig. 12a, red line). Note that we have also tested other ways to set the loss function weights (Supplementary Fig. 12a, orange and blue lines). The results show that the first weighting strategy produces the best forecast skills (Supplementary Fig. 12b).

### The attention blocks

Physical interpretability of the statistical models constructed with the artificial intelligence methods has been a long-standing difficulty. In order to increase the interpretability of the MTL-NET's results, we adopted the attention mechanism. It is well known that the attention plays an important role in human perception. Humans exploit a sequence of partial glimpses and selectively focus on salient parts in order to better capture visual structures. We inserted the channel attention block and spatial convolutional attention block[47] into the MTL-NET as follows.

$$Attention_{channel} = \sigma\left(W_1\left(W_0\left(F_{avg}^c\right)\right) + W_1\left(W_0\left(F_{max}^c\right)\right)\right) \qquad (2)$$

where $\sigma$ denotes the sigmoid function and $W_0$ and $W_1$ represent the weights of the model that are shared by both inputs (i.e., $F_{avg}^c$ and $F_{max}^c$). The ReLU activation function is followed by $W_0$, $F_{avg}^c$ and $F_{max}^c$ are obtained by aggregating the spatial information of a feature map using both the average-pooling and maximum-pooling operations that represent the average-pooled features and maximum-pooled features, respectively. Both descriptors are then forwarded to a dense layer to produce a channel attention map[47].

In addition, we also need a spatial attention map to capture the geographical importance of the predictors. We aggregated the channel information of a feature map by using two pooling operations, therefore generating two 2D maps: $F_{avg}^s$ and $F_{max}^s$. They represent the average-pooled features and maximum-pooled features across the channel, respectively. They are then concatenated and convolved by a standard convolutional layer, producing the 2D spatial attention map[47]. To clearly represent the key regions, the spatial attention map was standardized and values smaller than the average were ignored. In short, the spatial attention is computed as below.

$$Attention_{spatial} = \sigma\left(f^{n \times n}\left(\left[F_{avg}^s; F_{max}^s\right]\right)\right) \qquad (3)$$

where $\sigma$ denotes the sigmoid function and $f^{n \times n}$ represents a convolution operation with the filter size of $n \times n$. In the MTL-NET, we set n to 1 and use the 3D convolutional layer instead of the standard convolution.

### Sensitivity experiments

It is known that the prediction skill of machine learning models is dependent on different data. In order to test the possible overfitting problem and the sensitivity to the prediction test data in the MTL-NET, we designed three sensitivity experiments, in which we replaced the original wind and SST in test datasets with the NCEP/NCAR Reanalysis 1[48] and NOAA Optimum Interpolation Sea Surface Temperature V2[49], respectively. The prediction results from the sensitivity experiments show no significant difference from the original one, suggesting that our results are not sensitive to the selection of test dataset, and therefore the prediction results in the MTL-NET are robust (Supplementary Fig. 4).

## Data availability

Data related to this paper can be downloaded from: CMIP5 database, https://esgf-node.llnl.gov/search/cmip5/; CMIP6 database, https://esgf-node.llnl.gov/search/cmip6/; SODA, http://iridl.ldeo.columbia.edu/SOURCES/.CARTON-GIESE/.SODA/; NOAA-20Century Reanalysis version 3, https://psl.noaa.gov/data/gridded/data.20thC_ReanV3.monolevel.html; GODAS, https://psl.noaa.gov/data/gridded/data.godas.html; OISST, https://psl.noaa.gov/data/gridded/data.noaa.oisst.v2.html; NCEP/NCAR Reanalysis 1, https://psl.noaa.gov/data/gridded/data.ncep.reanalysis.html; NCEP-DOE Reanalysis 2, https://psl.noaa.gov/data/gridded/data.ncep.reanalysis2.html; NMME, http://iridl.ldeo.columbia.edu/SOURCES/.Models/.NMME/.

## Code availability

The deep learning models were developed using standard libraries in open-source platforms including Keras and TensorFlow. Codes used in this study are available from the corresponding author on request.

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

## Acknowledgements

J.-J.L. is supported by the National Key Research and Development Program of China (No. 2020YFA0608000) and National Natural Science Foundation of China (Grant 42030605).

## Author contributions

F.H.L. and J.-J.L. are co-first authors and wrote the manuscript. J.-J.L. conceived the idea of the study. F.H.L. designed the AI models and performed the MTL-NET hindcast experiments. F.H.L., Y.L. and T.T. performed the analysis under supervision of J.-J.L. F.H.L., J.-J.L., Y.L., T.T., L.B., W.O. and T.Y. contributed to interpreting results, discussions of associated dynamics and improvement of the presentation.

## Competing interests

The authors declare no competing interests.
