## [Peer Review File · Nature Communications]

Multi-task Machine Learning Improves Multi-Seasonal Prediction of the Indian Ocean DipoleReviewers' comments:

Reviewer #1 (Remarks to the Author):

This study applies a multi-task learning model to predicting the IOD and demonstrates skillful forecasts up to 8 months. Both the methodology and the results are potentially useful in improving the IOD prediction. However, some improvements are needed in analyzing and discussing the results. Therefore, I recommend a major revision.

Major comments

1. The authors may want to explain why they chose these variables (lines 66-69) as predictors. Are they obtained from observations?
2. Historical simulations of the CMIP5/6 models (Table S1) are used as the training data. Does the MTL-NET's skill in predicting the IOD depend on the CMIP models' performance in simulating the IOD? How many ensemble members are used for each CMIP model? When doing the hindcast over 1983-2019 for the IOD, does the training data exclude those over the hindcast period?
3. The authors claim that "the combination of the deep learning method with the geophysical big data can help deepen our understanding of complex climate variabilities in the Earth system, such as IOD events. Please highlight any new findings that have not been disclosed in previous studies for the IOD.
4. L20: The authors claim that the MTL-NET can "correctly interpret nonlinear dynamics of the IOD" in the abstract. However, the "nonlinearity" is not explicitly mentioned in their discussions.
5. L61-66: Primary task vs. secondary tasks. What are the differences between the primary and secondary tasks in terms of forecast skills? Should the skill in the primary task be better than those in the secondary tasks?
6. Figure 2 shows the comparisons between the MTL-NET and eight dynamical models in predicting IOD. How many models are currently used in operations? It would be better to compare the MTL-NET forecast skill with the models in major operational centers, such as the NCEP CFSv2 and those used in EC and China (CMA).
7. Figure 2 and Figure S3: Some discussions about the skills for IOD, WIO, and EIO are needed. Is the IOW (IOE; line 65) and WIOD (EIOD, line 88) the same thing? If so, what is the skill of IOD derived from the secondary tasks (IOW and IOE) as compared to that in the primary task (Fig. 2)?
8. L91-94: The statement is not supported by the data because not all state-of-the-art dynamical forecast systems are examined in this paper, especially the models in some major operational centers. I also noted that GFDL-aer04 has a better skill at the lead time of 11 months than the MTL-NET (Fig. 2a).

Minor comments and edits

1. L16: MTL-NET: What does "NET" stand for?
2. L15: "several months ahead" and L36: "a few months ahead": It would be better to be more specific (e.g., three months ahead).
3. I would suggest changing "automatically assess" to "help assess".
4. L43: Replace "where" by "which".
5. Figure S1c: Why do the forecasts show consistently large negative errors in positive IOD cases (marked with red stars)?
6. L81-84: The statement (Unlike) may not be true because the skills of some dynamical models are close to that of the MTL-NET, without a sharp decrease. Additionally, a forecast at "12-15 months" lead time is not a "multi-year" prediction.
7. L88: What does "increasing weights of the other tasks) mean?
8. Figure S4: Are the "original data" the CMIP historical simulations?
9. L104: At the 6-month lead, Vs is more important than HC (Fig. S5c).
10. What is the dash horizontal line in Fig. 2?
11. L120: Insert "." after "(Fig. 3a)".
12. L418: What "results" turn out to be robust?

Reviewer #2 (Remarks to the Author):

This study explored the IOD predictability using Multi-task deep learning framework. It was found that the MTL-NET can predict the IOD well up to several seasons ahead, outperforming all dynamical models. Further, the authors use the heat-maps from the MTL-NET to explore the predictability sources of the MTL-NET trained model. My comments are as below:

(1) The authors claimed that their MTL-NET can predict the IOD better than all world-class dynamical models. It seems not true since a recent work with a stochastic dynamical model has a very good skill for the fall IOD prediction (Zhao et al. 2020). On the other hand, there are numerous works to report that the machine learning defeated dynamical models for climate predictions (e.g., Ham et al. 2019). Even for the IOD itself, there were some reports to show much better skills by AI techniques than dynamical models, for example, Ratnam et al. 2020 and Liu et al. 2021 etc.. The authors should compare their predictions not only with GCM dynamical models but also some similar AI works.

(2) This work used the Multi-task framework which contains shared layers to predict both ENSO and IOD. The Multi-task neuron network is a popular framework to perform semantic segmentation or image pattern recognition. When applied to deal with language problems, we can feed the network with an article and then expect the network to do several tasks at the same time, for instance, write an abstract, translate into other languages and keywords analysis. One task will not confront the other. However, if we use a network to predict both ENSO and IOD, there might be a conflict between them. Moreover, it is totally wrong to manually adjust the Nino3.4 weight of the Loss function. The ENSO's power on the IOD, or other way around, can be exaggerated. And indeed, this happens in the study. In line 131 and 149, The authors indicate that the show up of abnormal area of the heatmap comes from the wrong interpretation of equatorial Pacific. If one key area can be misguided, how can we make sure that other regions are reliable? In my opinion, the precursors are infected by the false Pacific signal. The features obtained during ENSO-task training can easily destroy the features for IOD prediction. Although ENSO and IOD are highly related, the physical mechanism behind them must differ from each other. So, it would be more reasonable if the authors can exclude ENSO task from the MTL-NET and provide new result of skills and heatmaps.

(3) The heatmaps of Figures 3(a-b) and 4(a-b) are the values of spatial attention layer ($6*18*1$) resized at each grid point ($24*72*1$) (by the way, spatial attention doesn't change the size of feature map. So the orange block should have the same size of the dark blue block of conv3d). Therefore, the key regions are an overall perception of the three input variables. I feel difficult to know it is a SST key region or a HC key region. Also, the authors only gave the precursors at 3-month-lead and 12-month-lead. This might lead to false physical interpretation of the IOD onset. So, Is there any consistent signal? Given heat map as a function of lead time and target season, one can analyze the composite heat map of historical IOD events to see the whole process of precursor evolution. After all, it is entirely possible that the key region is just some superficial statistical information pathways not real physical pathways which may reside entirely somewhere else.

(4) Related to #2 and #3. Speaking of the precursor itself, the key region emerges only in the Pacific, even at short lead times. It seems to me not make sense. An Indian Ocean phenomena is totally at the mercy of the Pacific climate variability? Many studies indicate that some of IOD events, such as 1994 and 2006 events, are an internal mode that didn't co-occur with ENSO event (e.g., Behera et al 2006, Yang et al 2015). We believe that the Indian Ocean should contain some precursors in these two years at short leads. As commented above, the Multi-task may be blamed for this bias. It might also be adequate to divide the IOD events into two groups, namely the ENSO-forcing and internal mode, to study the heat maps.

(5) The authors should point out whether the result is an ensemble output or not. The deep learning (DL) method can be unstable to some extent. So, the result of one deep learning model is not reliable. The authors can use different number of neurons at each layer to build different DL models without changing its architecture (Ham et al 2019). Then a more complex ensemble strategy (Ham et al 2019) can be used to produce the final output.

(6) The authors only provided the SON skill in the main content of this study. It is widely known that the state of IOD peak season is relatively easier to predict when compared to that of other seasons (actually ENSO as well). Therefore, it is hard to evaluate the real IOD prediction improvement caused by MTL-NET. In fact, it seems that the MAM and JJA skills in supplementary information are not as excellent as the SON skill.

(7) The study didn't give forecasted timeseries of any target season at any lead times. The figure of forecasted timeseries is a quick and visual way to tell readers about your skills. More importantly, It also tells us the amplitude of your forecasted DMI, since the amplitude problem is a chronic disease in IOD prediction of machine learning (Ratnam et al 2020, Liu et al 2021). It will be much better if an all-season timeseries plot at different leads is shown.

(8) In Figure S5, how did this study quantify the contribution of different input? The channel attention is actually a SE block (Hu et al 2017). From what I know, there exists no reliable method to separate one input layer (e.g., SST) from another (e.g., HC), since the authors stack the 2D maps (SST, HC, winds) to a cube (like a real RGB map) as input. The relations between the three input variables are highly connected in both the feature map (extracted by convolution layer) and the final attention tensor. It is now a challenge even in the computer science. So, we would like to hear more details about it.

Here are some papers the authors should pay attention:

Jie Hu, Li Shen, Samuel Albanie, Gang Sun, Enhua Wu, 2017, Squeeze-and-Excitation Networks, <https://arxiv.org/abs/1709.01507>.

Behera, S. K., J. J. Luo, S. Masson, S. A. Rao, H. Sakuma, and T. Yamagata, 2006: A CGCM study on the interaction between IOD and ENSO. *J. Climate*, 19, 1688–1705, doi:10.1175/JCLI3797.1.

Yang, Y., Xie, S. P., Wu, L., Kosaka, Y., Lau, N. C., & Vecchi, G. A. (2015). Seasonality and predictability of the Indian Ocean Dipole mode: ENSO forcing and internal variability. *Journal of Climate*, 28(20), 8021–8036. <https://doi.org/10.1175/JCLI-D-15-0078.1>

Zhao et al, 2020: Improved Predictability of the Indian Ocean Dipole Using a Stochastic Dynamical Model Compared to the North American Multimodel Ensemble Forecast, *J Climate*, DOI: 10.1175/WAF-D-19-0184.1, Vol 35,

Ratnam et al., 2020: A machine learning based prediction system for the Indian ocean dipole, <https://doi.org/10.1038/s41598-019-57162-8>,

Liu et al, 2021: Forecasting the Indian Ocean Dipole with deep learning techniques. *Geophys. Res. Lett.* 48, e2021GL094407 228

Response to Reviewer #1

We appreciate all valuable comments/suggestions from the reviewer. All comments have been carefully addressed and the manuscript has been revised accordingly.

Major comments:

1. The authors may want to explain why they chose these variables (lines 66-69) as predictors. Are they obtained from observations?

Response: The reason we chose these variables as predictors is that the Indian Ocean Dipole (IOD) is an intrinsic ocean-atmosphere coupled process. Such that, not only the oceanic signal needs to be considered, but that from atmosphere is also crucial. The data shown in Table S2 are not only from observations but also derived from CMIP model outputs and three different reanalyses dataset (i.e., 20th Century, GODAS and NCEP Reanalyses), respectively. We have clarified this in the revised manuscript (see Line 85-89).

2. Historical simulations of the CMIP5/6 models (Table S1) are used as the training data. Does the MTL-NET's skill in predicting the IOD depend on the CMIP models' performance in simulating the IOD? How many ensemble members are used for each CMIP model? When doing the hindcast over 1983-2019 for the IOD, does the training data exclude those over the hindcast period?

Response: Thank you for your comments. Since the deep machine learning model requires big training data, we have used as many model outputs as we could download online when this work was started. We have added the column in Table S1 to indicate how many ensemble members used for each model. Please note that most of CMIP5 and CMIP6 models tend to overestimate the IOD intensity (see Fig. R1), which is a well-recognized systematic bias. Concerning this bias, we have adopted all models including a small number of models with underestimated IOD (see Fig. R2) in order to reduce the potential influence of the overestimation bias. Following the similar reason

considered in Ham et al. (2019), we did not attempt to select CMIP models for the training, because the IOD is influenced by complicated air-sea coupled processes and it is hard to objectively decide which model is really better than the others based on the evaluation of existing indices. For instance, a better simulation of the IOD index itself does not guarantee the model is better, simply because the better IOD index can be resulted from error cancellations between worse physical processes.

As shown in Table S2, the training dataset using CMIP5/6 historical outputs spans from 1860 to 2009, and the test dataset is from GODAS and NCEP Reanalysis products with the period of 1983-2019. Since the model historical simulations do not reproduce the interannual variations of observed IOD, they are independent from the observed records. Therefore, we kept the training period as long as possible (see also Ham et al. 2019).

Figure R1. The standard deviations of the IOD index. Three-month running average of standard deviations of the IOD index based on observation (black line), multi-model ensemble means of CMIP5 (blue line) and CMIP6 (red line) models. The blue and red shadings indicate the interquartile range of CMIP5 and

CMIP6 models.

Figure R2. The standard deviations of the IOD index in different models.

Similar to Figure R1, but for individual models in CMIP5 and CMIP6.

3. The authors claim that “the combination of the deep learning method with the geophysical big data can help deepen our understanding of complex climate variabilities in the Earth system, such as IOD events. Please highlight any new findings that have not been disclosed in previous studies for the IOD.

Response: Our results revealed a possible new mechanism that how the sea surface temperature (SST) in the North Pacific Ocean influences the development of a strong positive IOD event. As shown in Fig. 4a, the SST warming in the North Pacific occurred 12-month in advance can be captured by our deep learning prediction model. In a recently accepted publication by Ding et al. (2022), it shows that such a warming in a 12-month lead time affects the equatorial eastern-central Pacific through the mechanism of seasonal footprint and induces a multi-year El Niño event. As widely recognized, El Niño can subsequently promote the occurrence of a positive IOD. Thus, the finding of Ding et al. (2022) supports our deep learning model results. Furthermore, our model also reveals distinctive 12-month lead precursors responsible for the occurrence of positive and negative IOD events. This helps deepen the understanding of the nonlinear mechanisms of IOD. We have highlight this in the revised manuscript (see Line 215-218).

4. L20: The authors claim that the MTL-NET can “correctly interpret nonlinear dynamics of the IOD” in the abstract. However, the “nonlinearity” is not explicitly mentioned in their discussions.

Response: “nonlinear dynamics of the IOD” means that the mechanisms responsible for positive and negative IOD events are distinctive, as was shown in Figs. 3 and 4, and Figs. S8-S10. The MTL-NET model is built on a convolution neural network (CNN) with embedded spatial and channel attention blocks. In CNN, the activation function is applied on the weighted sum of the input and transformed into an output from nodes in each CNN layer. The activation functions must be nonlinear. It helps to improve the efficiency of deep learning. Thus, the predicted IOD events in our model are the results obtained from a series of nonlinear transformation. We have rephrased the sentence as “the MTL-NET can help assess the importance of different predictors and correctly capture the nonlinear relationship between the IOD and predictors.” (see Line 25-27). We have also added briefly discussions as follows (see Line 230-241 in the revised manuscript).

“Our study suggests that deep learning methods can extend reliable IOD predictions out to 7 months ahead, and the combination of the deep learning method with the geophysical big data can help deepen our understanding of complex climate variabilities in the Earth system. Traditional analyses largely rely on linear regression to explore the relationship between the predictors and predictands and the nonlinear relationships are often neglected. The deep learning method can be an effective complementary as the deep learning model is built on a series of nonlinear calculations. Based on the interpretable analyses on our model results, we can obtain the distinctive mechanisms responsible for positive and negative IOD events; this helps deepen the understanding of the nonlinear mechanisms of IOD, although they need to be fully tested by traditional dynamical model experiments and other approaches.”

5. L61-66: Primary task vs. secondary tasks. What are the differences between the primary and secondary tasks in terms of forecast skills? Should the skill in the

primary task be better than those in the secondary tasks?

Response: The difference between the primary and secondary tasks is their weights applied on the neurons. The weights in the primary task are larger than those in secondary task, which can lead to larger loss in the primary task. The model tends to capture more features or decrease the larger loss with deepening of the network. Yes, primary skills are better than secondary skills in our model design, because by placing larger weights on the loss function, the model will extract more features in the primary task, resulting in better skills. To demonstrate this, we have conducted two additional experiments by swapping the primary and secondary prediction targets. In the experiment of predicting WIOD as a primary task, the EIOD, DMI, and Nino3.4 are secondary tasks, while in the one with EIOD as a primary task, the rest three indices become secondary tasks. Fig. S3 shows that the predictions of WIOD and EIOD as primary tasks also show higher skills than most of 14 dynamical model forecast systems, and Fig. R3 show the prediction skills in the primary task are better than those in the secondary tasks. We have added Fig. S11 (i.e., Fig. R3) and briefly explained skill differences between primary and secondary tasks in Methods (see Line 462-465).

Figure R3. Prediction skills for SON DMI based on MTL-NET with different predictand as primary task. Predictive skill of the SON DMI as a function of the forecast lead month in the MTL-NET with DMI as primary task (red line), MTL-NET with EIOD as primary task (blue line), and MTL-NET with WIOD as primary task

(orange line). The validation period is from 1983 to 2019.

6. Figure 2 shows the comparisons between the MTL-NET and eight dynamical models in predicting IOD. How many models are currently used in operations? It would be better to compare the MTL-NET forecast skill with the models in major operational centers, such as the NCEP CFSv2 and those used in EC and China (CMA).

Response: Thanks for the advice. In this study, we compared 7 dynamical models from the North American Multi-Model Ensemble (NMME) and SINTEX-F model (which is in operation, see <https://icar.nuist.edu.cn/en/111/list.html> for monthly updated forecast products). We would like to add more models for skill comparison, but international data access is rather restricted, particularly in China, and we did not get CMA model outputs due to their data policy. Another reason is that the hindcast period of many operational forecast models such as ECMWF and NCEP (see <https://cds.climate.copernicus.eu/cdsapp#!/dataset/seasonal-monthly-single-levels?tab=form>) is limited to the period after 1993, shorter than that of NMME and SINTEX-F (i.e., 1983-2019). In addition, Shi et al. (2012) have already compared operational models in predicting IOD, including NCEP CFSv1 and v2, ECsys3, two POAMA versions, and SINTEX-F. Their results showed that the prediction skills of the SON DMI are generally comparable out to 6-month lead and SINTEX-F's skill is slightly higher than the other operational models at 5 and 6-months lead (see Fig. 5a of Shi et al. 2012). Therefore, compared with the results shown in Shi et al. (2012), the prediction skill of our MTL-NET model is also higher than the skills of these operational models. In order to better address the reviewer's questions, we spent several months to downloading the hindcast results of the six operational models including EC, NCEP, and compare the MTL-NET forecast skill with these models (Fig. R4). The prediction skill is validated for the common period of 1993-2016. The results show that our MTL-NET ranked top 2 in predictive skill out to 5 months lead. For a better comparison, we have adopted these 14 dynamical models' hindcasts and revised Fig. 2

(i.e., Fig. R5). We have briefly highlighted this in the revised manuscript (see Line 98-100).

Figure R4. Correlation skills of the IOD prediction. (a) Predictive skill of the SON DMI as a function of the lead month based on the MTL-NET (red line), six operational forecast models which the hindcast periods are limited to the period after 1993 (the other coloured lines). The prediction skill is validated for the period of 1993-2016. Black dashed line denotes the skill of 0.5. (b) Root mean square error (RMSE) of the SON DMI prediction based on the MTL-NET and each dynamical model forecast system.

Figure R5. Correlation skills of the IOD prediction. (a) Predictive skill of the SON DMI as a function of the lead month based on the MTL-NET (red line), SINTEX-F dynamical forecast system (blue line), six operational forecast models which the hindcast periods are limited to the period after 1993 and seven dynamical forecast systems of the North American Multi-Model Ensemble (NMME) project (the other coloured lines). The prediction skill is validated for the period of 1983-2019. Black dashed line denotes the skill of 0.5. (b) Root mean square error (RMSE) of the SON DMI prediction based on the MTL-NET and each dynamical model forecast system. The diagonal line represents the model does not have prediction at this lead time. (c) The DMI in Sep-Oct-Nov (SON) based on the observations (black line) and 10-member ensemble mean predictions of MTL-NET at lead time of 1, 3, 6 and 12 months (red, blue, orange and green dashed lines), respectively.

7. Figure 2 and Figure S3: Some discussions about the skills for IOD, WIO, and EIO are needed. Is the IOW (IOE; line 65) and WIOD (EIOD, line 88) the same thing?

If so, what is the skill of IOD derived from the secondary tasks (IOW and IOE) as compared to that in the primary task (Fig. 2)?

Response: Yes, the IOW (IOE) and WIOD (EIOD) are the same thing. We have removed this inconsistency and used “EIOD” and “WIOD” throughout the manuscript and added brief discussions about the three indices. Prediction skills of the IOD from the WIOD and EIOD are obtained in two ways: (1) The skill of the IOD derived from the secondary task (i.e., WIOD minus EIOD); (2) The skill of the IOD derived from WIOD and EIOD when setting them as the primary task (Fig. S3). We compare them with the skill of the IOD in the primary task (Fig. R6). As expected, the skills of the IOD prediction calculated using the EIOD and WIOD indices is lower than the skill of the IOD as a primary task shown in Fig. 2. The result proves the effectiveness of the multi-task framework. We have added these skills in Fig. S1 and brief discussions in Methods (see Line 465-468).

Figure R6. The correlation skills of the IOD prediction. Prediction skills of the SON DMI as a function of the lead month from the primary task (red line, see Fig. 2), that derived from the secondary tasks (i.e., WIOD minus EIOD, blue line) and that derived from WIOD and EIOD when set them as primary task in MTL-NET (Figure S3, orange line).

8. L91-94: The statement is not supported by the data because not all state-of-the-art dynamical forecast systems are examined in this paper, especially the models in some major operational centers. I also noted that GFDL-aer04 has a better skill at the lead time of 11 months than the MTL-NET (Fig. 2a).

Response: Thanks for this comment. We have rephrased this statement to make it more accurate (see Line 119-122).

“Therefore, we conclude that the MTL-NET provides a reliable forecast of the IOD events up to 7 months ahead and even beyond, which has yet not been achieved in most of the fourteen dynamical forecast systems.”

Minor comments and edits

1. L16: MTL-NET: What does “NET” stand for?

Response: The “NET” is not an acronym. It simply means “net” literally. Following the conventional naming method for a deep learning model, the “NET” represents any model with enormous nodes connected with each other with a lot of weights, acting like a network.

2. L15: “several months ahead” and L36: “a few months ahead”: It would be better to be more specific (e.g., three months ahead).

Response: Yes, we have changed in the manuscript as follows (see Line 22-27, 48-50).

“Hindcasts of the IOD events during the past four decades indicate that the MTL-NET can predict the IOD well up to 7-month ahead, outperforming most of world-class dynamical models used for comparison in this study. Moreover, the MTL-NET can help assess the importance of different predictors and correctly capture the nonlinear relationship between the IOD and predictors.”

“the prediction of the IOD events (except for the super IOD event in 2019) is still limited to three months ahead in many current state-of-the-art climate model forecast systems.”

3. I would suggest changing “automatically assess” to ‘help assess”.

Response: Thank you for your suggestion. We have changed it in the manuscript (Line 25).

4. L43: Replace “where” by “which”.

Response: We have amended on request (Line 47).

5. Figure S1c: Why do the forecasts show consistently large negative errors in positive IOD cases (marked with red stars)?

Response: Since our predictions are based on the average of 10 machine learning models, the ensemble mean always tends to underestimate the intensity of the IOD. Note that considerable spread exists among the 10 members (Fig. R7); this suggests that the intensity of observed IOD events may be better captured in some members occasionally. As the reviewer pointed out, the prediction error of positive IOD events appears to be larger than that of negative IOD events. This is probably because the intensity of positive IOD events is usually larger than that of negative IOD cases; this is a well-known asymmetry of the IOD (e.g., Hong et al. 2008). Note that the forecast error of extreme IOD events appears to be much larger. This is because the occurrence number of extreme events in history is very small and the mechanisms of extreme

events are not well understood; it is generally more difficult to make accurate forecasts of extreme events.

Figure R7. Timeseries of predicted DMI at 1-month lead. The shading denotes the range between the minimum and maximum values among the 10 ensemble members of the MTL-NET predictions.

6. L81-84: The statement (Unlike ...) may not be true because the skills of some dynamical models are close to that of the MTL-NET, without a sharp decrease. Additionally, a forecast at “12-15 months” lead time is not a “multi-year” prediction.

Response: We have rewritten this statement as follows (see Line 103-109 in the revised manuscript).

“The prediction skills of the DMI based on 13 out of 14 dynamical models are lower than the MTL-NET at 1-month lead. With the increase of lead time, the skills of most dynamical models decrease rapidly. Although some dynamical models have comparable skills at 1-4 months lead, their prediction skills become lower than that of the MTL-NET afterwards. It is worth noting that the MTL-NET can predict SON DMI at lead time beyond 12 months with a correlation skill close to 0.4 up to 15-month lead.”

7. L88: What does “increasing weights of the other tasks mean?”

Response: The primary and secondary tasks can be set in the MTL-framework by modifying weights to meet different demands. Increasing weights of the other secondary tasks can help extract the information of these tasks, which can in turn help these task to fit well, thus generating relatively better prediction results (see Methods, Line 456-465). We have rewritten this statement as follows (see Line 115-118).

“As an advantage of the multi-task framework, if setting the other tasks (e.g., the EIOD and WIOD indices) as the primary task, it can simultaneously produce their predictions that are also superior to most of the fourteen dynamical models’ performance (Fig. S3).”

8. Figure S4: Are the “original data” the CMIP historical simulations?

Response: The “original data” refers to the original test dataset. Here, we conducted some sensitivity experiments by replacing one of the variables in the original test dataset by that from different sources. For example, the SST data in the original test data is from the GODAS, while it is replaced by the NOAA Optimum Interpolation SST v2 in the “changing SST” sensitivity experiment. We have clarified this in Methods and Fig. S4 (see Line 514-522).

9. L104: At the 6-month lead, Vs is more important than HC (Fig. S5c).

Response: Yes, Vs appears to be the most important predictor among the four predictors at 6-month lead. However, the importance of HC is quite comparable to that of Vs. Here, we want to emphasize that the importance of HC, relative to that of the other predictors, is stably increasing with the increase of lead time. This is different from the varying importance of the other predictors among the different lead times. To clarify this, we have rephased this sentence in the manuscript as follows (see Line 132-135).

“As expected, the importance of the heat content anomalies of the upper ocean to the IOD prediction becomes more remarkable with the increase of lead time due to strong oceanic memory⁹. This is different from the varying importance of the other predictors

among different lead times.”

10. What is the dash horizontal line in Fig. 2?

Response: The dash horizontal line in Fig. 2 marks when the correlation skill equals to 0.5, which is often considered as a useful prediction in existing literature. We have clarified this in the caption of Fig. 2a.

11. L120: Insert “.” After “(Fig. 3a)”.

Response: We have amended on request.

12. L418: What “results” turn out to be robust?

Response: Some trained deep learning models tend to be overfitted or the predicted results from these models are dependent on the test dataset. It is important to test whether the predicted results from the MTL-NET model are sensitive to the test data that are selected. If the prediction skills of IOD are not affected by the selection of test dataset, we consider the predicted results are robust. As shown in Fig. S4, the skills in predicting IOD are not affected by the selection of different test dataset, so here we concluded the results turn out to be robust. We have briefly explained this in Methods (see Line 514-522).

Reference:

1. Ding, R., et al. Multi-year El Niño events tied to the North Pacific Oscillation. *Nat. Commun.* **13**, 1-11 (2022), <https://doi.org/10.1038/s41467-022-31516-9>.
2. Liu, J., Tang, Y., Wu, Y., Li, T., Wang, Q., Chen, D. Forecasting the Indian Ocean Dipole with deep learning techniques. *Geophys. Res. Lett.* **48**, e2021GL094407 (2021).
3. Ham, Y. G., Kim, J. H. Luo, J.-J. Deep learning for multi-year ENSO forecasts. *Nature* **573**, 568–572 (2019).
4. Hong, C. C., Li, T., Ho, L., Kug, J. S. Asymmetry of the Indian Ocean Dipole. Part

- I: observational analysis. *J. Clim.* **21**, 4834–4848 (2008).
5. Shi, L., et al. How predictable is the Indian Ocean Dipole? *Mon. Weather Rev.* **140**, 3867–3884 (2012).

Response to Reviewer #2

We appreciate all valuable comments/suggestions from the reviewer. All comments have been carefully addressed and the manuscript has been revised accordingly.

Major comments:

1. The authors claimed that their MTL-NET can predict the IOD better than all world-class dynamical models. It seems not true since a recent work with a stochastic dynamical model has a very good skill for the fall IOD prediction (Zhao et al. 2020). On the other hand, there are numerous works to report that the machine learning defeated dynamical models for climate predictions (e.g., Ham et al. 2019). Even for the IOD itself, there were some reports to show much better skills by AI techniques than dynamical models, for example, Ratnam et al. 2020 and Liu et al. 2021 etc. The authors should compare their predictions not only with GCM dynamical models but also some similar AI works.

Response: Thanks for the comments. We would like to clarify that the dynamical model here is referred to a global ocean-atmosphere fully coupled model, which differs from Zhao et al's statistical model (although they called it “stochastic dynamical model”). Zhao et al. (2020) used the training period of 1982-98 for estimating all the parameters of their multiple regression model, which was then applied to predict the IOD during 1982-2018 (so there is partly overlapping period). We made a rough comparison based on the results shown in Zhao et al's article, which appears to have a higher forecast skill than our model. We have toned down the statement to make it more accurate (see Line 22-25, 119-122).

We would like to add more dynamical model and AI models for skill comparison, but it is difficult to rebuild these models described in other publications because details of their datasets and models are not available for downloading. Another reason is that the initial forecast time and the length of the forecast are not consistent among the publications. Nevertheless, to best address the reviewer's comments, we tried to

compare our model with the ones that are similar in predicting timeframe (i.e., Liu et al. 2021). Fig. R8 shows the prediction skills as a function of initial season (month) and target month based on the CNN model of Liu et al. (2021). They used the predictors averaged in three months to predict the DMI in the following month and marked it as the forecast at lead time of 1-month. For example, when predictors in Jun-Jul-Aug (JJA) were used to predict the DMI in September, they defined it as 1-month lead forecast. The peak IOD occurs during Sep-Oct-Nov (SON). Therefore, in their study, the 1-month lead forecast for SON DMI is the averaged predictions from the Jun-Jul-Aug (JJA), Jul-Aug-Sep (JAS), and Aug-Sep-Oct (ASO). Their technique is different from ours, as the MTL-NET only use JAS to predict the DMI in SON (i.e., 1-month lead forecast in our definition). For a fair comparison, we converted our lead time to match theirs before comparing (Fig. R9). As the results show, if we follow the way of their article, our skill is slightly better than theirs at 7 months lead. This comparison was added to the discussion in Methods (see Line 447-449).

Figure R8. Prediction skills of IOD in the CNN model proposed by Liu et al. (2021). The correlation skill as a function of the initial season and target month in their CNN model (i.e., their Fig. S1).

Figure R9. The predicted DMI in Sep-Oct-Nov (SON) at lead time of 1, 4, and 7 months based on (a) our MTL-NET and (b) the CNN model of Liu et al., 2021 (i.e., their Fig. 3).

2. This work used the multi-task framework which contains shared layers to predict both ENSO and IOD. The Multi-task neuron network is a popular framework to perform semantic segmentation or image pattern recognition. When applied to deal with language problems, we can feed the network with an article and then expect the network to do several tasks at the same time, for instance, write an abstract, translate into other languages and keywords analysis. One task will not confront the other. However, if we use a network to predict both ENSO and IOD, there might be a conflict between them. Moreover, it is totally wrong to manually adjust the Nino3.4 weight of the Loss function. The ENSO's power on the IOD, or other way around, can be exaggerated. And indeed, this happens in the study. In line 131 and 149, The

authors indicate that the show up of abnormal area of the heatmap comes from the wrong interpretation of equatorial Pacific. If one key area can be misguided, how can we make sure that other regions are reliable? In my opinion, the precursors are infected by the false Pacific signal. The features obtained during ENSO-task training can easily destroy the features for IOD prediction. Although ENSO and IOD are highly related, the physical mechanism behind them must differ from each other. So, it would be more reasonable if the authors can exclude ENSO task from the MTL-NET and provide new result of skills and heatmaps.

Response: Thank you for your comments. We have invited two AI experts, Dr. Bai and Dr. Ouyang, from the University of Sydney, to help examine all details of our MTL-NET and to make sure all AI aspects are correct. The responses to each comment are given as follows.

Q1: The issue about the conflict between ENSO and the IOD.

As the reviewer acknowledged that ENSO and IOD are highly related, they should not be considered to be in conflict with each other. This is explained below from both the physical and the machine learning perspective.

It is well-known that ENSO in the tropical Pacific Ocean and the IOD in the Indian Ocean are highly coupled with each other via atmosphere and ocean bridges (see Cai et al. 2019 for a recent review published in *Science*, references therein). While some IOD events can happen in absence of ENSO owing to internal air-sea coupled processes in the tropical Indian Ocean, the co-occurrence with ENSO can largely promote the development of IOD. *Vice versa*, the IOD can also play an important role in the development of ENSO. Therefore, ENSO and IOD must be viewed in an inter-basin coupled system of the Pacific and Indian Ocean. The inter-basin coupling among the tropical oceans has been widely recognized and considered to be an important advance to understanding the tropical climate variations. Different from the previous single-task model that ignores the important inter-basin interactions, our MTL-NET is design to

simulate the interactions between ENSO and the IOD, and hence produces better forecasts of the IOD. We have briefly explained this in Methods (see Line 73-78, 427-434, 442-444, and Fig. S1).

From the perspective of machine learning and data mining, regarding a single-task learning framework, perhaps the filters in the convolutional layer would mostly be used to extract the Indian Ocean (or the Pacific Ocean) signal due to the principle of least squares, to ensure optimal results are obtained. But this is not consistent with the dynamics of the phenomenon in the real world, as was described above. Hence the multi-task learning framework were designed to solve this problem. In a multitasking framework, the choice of auxiliary tasks is important and is very relevant to the performance of the model. For instance, Bingel et al. (2017) analysed why a multi-task learning framework (MTL) works and highlighted that MTL works well when the primary task is harder to learn and the auxiliary task is easier to learn. The auxiliary task is more like a constraint on the primary task, in order to make the primary task learn more comprehensive features and better match the high-level non-linear features of the input. Of course, the above method assumes that the main and auxiliary tasks are relevant, and the input data are shared. If one uses some pictures of cat to classify the colours of cats and predict the DMI, this is very unreasonable as the two tasks are not related and the data is not shared, the cat pictures do not have any IOD information. In our MTL model containing the IOD in the Indian Ocean and ENSO in the Pacific, the two climate signals are highly related due to the strong Indo-Pacific inter-basin coupling. And since ENSO (i.e., the secondary task) has much higher predictability than that of the IOD, our designed MTL can obtain better predictions of IOD with constraints from the Pacific Ocean features when parameters are shared across multiple tasks and features are shared across the tasks. In summary, there is no conflict between ENSO and the IOD regarding the MTL method itself. And it has been documented in many previous studies that different tasks in MTL can be worked as a complementary to each other (e.g., Kendall et al. 2018; Liu et al. 2019). We also briefly explained this in

Methods (see Line 456-462).

Q2: The issue about manual adjustment on the weights of the loss function.

We would like to clarify that it is not wrong to adjust the weights of the loss function manually.

Firstly, it has been widely known that the tasks use a naive weighted sum of losses, where the loss weights can be either uniform or manually tuned in MTL, although the manual adjustment are dependent on physical knowledge and experience. (e.g., Sermanet et al. 2014; Eigen et al. 2015; Kokkinos et al. 2016).

Secondly, some previous models have already demonstrated that setting the loss function weights helps obtain best predictions, such as the way based on the variance contribution obtained from the EOF analysis, weighted loss of precipitation prediction, Cycle GAN and RADA, etc (e.g., Higgins et al. 2017; Lin et al. 2017; Zhu et al. 2017; Ravuri et al. 2021; Pan et al. 2021). These previous results all support and justify the way of manually setting weights of loss functions.

Finally, previous studies have shown that, with the increase of lead time, the role of the Pacific Ocean in influencing the IOD may become more and more important, and hence we can obtain more features about the Pacific Ocean by increasing the weights of Nino3.4 task based on the premise that the main task is the prediction of the IOD in our model. In our opinion, setting the weights of the loss function in a way that takes into account the Indo-Pacific inter-basin interactions is well in line with the physical knowledge and is a worthy and innovative attempt.

We set the weight of the DMI task to 5 and the weights of the other tasks to 1 at 1-month lead, and allow the weight of Nino3.4 task gradually increases with the lead time (Fig. R10a, red line). In addition, we have also tested other ways to set the loss function weights (Fig. R10a, orange and blue lines). The results show that our current weighting strategy produces the best forecast skills (Fig. R10b). We have briefly explained this in

Methods (see Line 482-486 and Fig. S12).

Figure R10. Prediction skills based on different weights set for the ENSO task in the loss function. (a) The different weights set for the ENSO task in the loss function. (b) The correlation skill of IOD prediction based on the different ways of setting the weights in the loss function. The validation period is from 1983 to 2019.

Q3: The issue about the heatmap with wrong interpretation of equatorial Pacific.

We disagree with this comment. We believe that the interpretation of the equatorial Pacific signal is correct and consistent with the well-known physics. As was described in the above responses to Q1 and Q2, the Pacific and Indian Oceans have strong interactions between them. And the importance of the equatorial Pacific signals to the

IOD development has been widely recognized in existing literature (see Cai et al. 2019 for a review, abundant references therein). Note that we had difficulty to understand the signals in the Arabian Sea. To address the reviewer's concern, we carefully re-examined this issue and found that the importance of the Arabian Sea signal is also consistent with the existing physical understanding (see Line 182-189).

Q4: The issue about the ENSO's influence in the MTL-NET.

We understand the reviewer's concern on the influence of ENSO on the prediction of IOD. We designed an additional experiment with the same parameters but removing the influence of ENSO. The results are shown as follows.

Firstly, the ACC skills of MTL-NET without the Nino3.4 index is similar to MTL-NET at 1- to 4-month lead. But the skill drops at 5-month lead and beyond, becomes much lower than the MTL-NET (Fig. R11). As for the importance of the predictors revealed by the channel attention and the heat map, the results are generally similar (cf. Fig. R12 and S5; Figs. R13 and 3-4). However, the results based on the MTL-NET with ENSO task provides clearer and more area-focused information on the importance of the predictors at different lead times. The results support the advantage of our MTL-NET. To clarify this, we have added the skill based on the MTL-NET without ENSO task in Fig. S1 and brief discussions in Methods (see Line 475-481).

Figure R11. Prediction skills of the IOD during SON. Correlation skills of the SON DMI as a function of lead month based on the MTL-NET (red line) and the MTL-NET without Nino3.4 index (blue line)

Figure R12. The importance of the predictors estimated by the MTL-NET in the experiment without the influence of ENSO at different lead times. The results are produced by the channel attention mechanism for the SON DMI predictions initiated from (a) Jul-Aug-Sep (JAS), (b) May-Jun-Jul (MJJ), (c) Feb-Mar-Apr (FMA), and (d)

Aug-Sep-Oct (ASO (-1)) in previous year, respectively.

Figure R13. The heat maps of the IOD predictions at 3- and 12-month lead in the experiment without the influence of ENSO. (a) Spatial attention map of the five strong pIOD events during 1983-2019 based on the 3-month lead predictions, i.e., initiated from May-June-July (MJJ). (c) Spatial attention map of the five strong pIOD events during 1983-2019 based on the 12-month lead predictions, i.e., initiated from August-September-October of the previous year (ASO (-1)). (b, d,) As in (a, c), but for the results of the five strong nIOD events.

3. The heatmaps of Figures 3(a-b) and 4(a-b) are the values of spatial attention layer ($6 \times 18 \times 1$) resized at each grid point ($24 \times 72 \times 1$) (by the way, spatial attention doesn't change the size of feature map. So the orange block should have the same size of the dark blue block of conv3d). Therefore, the key regions are an overall perception of the three input variables. I feel difficult to know it is a SST key region or a HC key region. Also, the authors only gave the precursors at 3-month-lead and 12-month-lead. This might lead to false physical interpretation of the IOD onset. So, is there any consistent signal? Given heat map as a function of lead time and target season, one can analyze the composite heat map of historical IOD events to see the whole process of precursor evolution. After all, it is entirely possible that the key region is

just some superficial statistical information pathways not real physical pathways which may reside entirely somewhere else.

Response: Thanks for the comments. We have revised Figure 1 (repeated here as Figure R14) to clarify the confusion. As the reviewer pointed out, the heat map provides the overall importance of the four predictors at different lead times, it does not show the separated role of each predictor. This may be one of issues to be improved for all similar type of deep learning models. Note that this defect may be somewhat improved by examining the importance of each predictor revealed by the attention channel (e.g., Fig. S5). It is also worth noting that the IOD is generated by ocean-atmosphere coupled processes, and hence it may not be necessary to split the roles of individual predictors since they highly interact with one another. Knowing the key regions of their overall importance can also provide useful information on the possible precursors of the IOD.

To verify whether the information revealed by the heat maps are reasonable or not, we have employed a statistical analysis on different predictors and explained how these precursors affect the IOD at different lead months (see Figs. 3 and 4). They are generally consistent with current understanding of the IOD mechanisms. We have also examined 1-year evolutionary map of the strong IOD events (Fig. S8-10). The results also support the heatmaps at 3- and 12-month lead. Therefore, we believe that there are indeed consistent signals. Due to the space limit, we cannot display all heatmaps at every lead time. We have briefly explained the reason why we chose 3-month and 12-month lead for interpretable analysis (see Line 139-142 and 190-192). As mentioned in the manuscript, the 3-month lead represents the current level of the IOD prediction skill in many ocean-atmosphere coupled dynamical model forecast systems, and hence it is interesting and important to explore the possible precursors at 3-month lead. In addition, as shown in Fig. 2, our MTL-NET and a few dynamical models display a skill rebound at 10-12 months lead. We believe this is an interesting phenomenon and may indicate some multi-seasonal predictability of the IOD. Exploring possible precursors at 12-month lead may help improve the understanding and multi-seasonal prediction of the

IOD.

The above results indicate that MTL-NET can discover some useful information that helps better understand IOD events. We generally agree with the reviewer that all similar type of deep learning models do not automatically guarantee that the results are accurate and consistent with the real physics, and we also mentioned that dynamical model sensitivity experiments and other ways may be required to verify whether the signals revealed by the deep learning model are true or not (see Line 237-243). Note that a recently accepted paper of Ding et al. (i.e., “Multi-year El Niño events tried to the North Pacific Oscillation” in Nature Communications) shows that the signal in the North Pacific at 12-month lead can affect the equatorial eastern-central Pacific through the seasonal footprint mechanism and induces a multi-year El Niño event. As was widely recognized, El Niño can subsequently promote the occurrence of a positive Indian Ocean Dipole. Thus, the finding of Ding et al. (2022) supports our deep learning model results. Furthermore, our MTL-NET model also reveals distinctive 12-month lead precursors responsible for the occurrence of positive and negative IOD events. This helps deepen the understanding of the nonlinear mechanisms of IOD. We have briefly added the above discussions.

Figure R14. The structure of the MTL-NET.

4. Related to #2 and #3. Speaking of the precursor itself, the key region emerges only

in the Pacific, even at short lead times. It seems to me not make sense. An Indian Ocean phenomenon is totally at the mercy of the Pacific climate variability? Many studies indicate that some of IOD events, such as 1994 and 2006 events, are an internal mode that didn't co-occur with ENSO event (e.g., Behera et al 2006, Yang et al 2015). We believe that the Indian Ocean should contain some precursors in these two years at short leads. As commented above, the multi-task may be blamed for this bias. It might also be adequate to divide the IOD events into two groups, namely the ENSO-forcing and internal mode, to study the heat maps.

Response: Thanks for the comments. We agree with the reviewer that the IOD sometimes can happen without ENSO's influence. However, we cannot agree with the point that the multi-task model must be blamed for the bias. First, the occurrence of independent IOD events is rather rare. Their information may not be statistically significant and hence does not show up in the heat maps with nonsignificant signals being omitted. Note that the IOD events in 1994 and 2006 actually co-occurred with central-Pacific type of El Niño, they are not purely independent. The heat maps tend to indicate the strongest precursors at 3-month lead, which is consistent with the composite signals shown in Fig. 3c. The anomalies of winds, SST and upper ocean heat contents in the equatorial Pacific at 3-month lead are indeed stronger than those in the tropical Indian Ocean. Therefore, the heat maps produced by the MTL-NET are realistic. In addition, it is worth noting that the heat map for the negative IOD events at 3-month lead displays statistically significant signals in the Indian Ocean, which is also consistent with the stronger signals there shown in the composite map (Fig. 3b and 3d). Thus, the heat maps do provide statistically significant, realistic and distinctive precursors for the positive and negative IOD events.

We also examined the 3-month lead heatmaps of the two IOD events (i.e., 1994 and 2006) that the reviewer mentioned (Fig. R15). Again, the key regions of the precursors revealed by the heat maps are also consistent with the observed signals. Note that the key regions now contain the tropical Indian Ocean, particularly in 1994 case. The

distinctive heatmaps between the 1994 and 2006 cases indicate that the MTL-NET can distinguish the different precursors for individual IOD events.

Figure R15. The physical interpretation of the IOD in 1994 and 2006. (a, d) The heatmaps of the positive IOD predictions at 3-month lead (i.e., MJJ) in 1994 and 2006, respectively. Only the values significant at the 90% confidence level based on 10 ensemble members are displayed. (b, e) The observed anomalies of SST (shaded) and the 10-m wind (vector) during MJJ in 1994 and 2006, respectively. (c, f) The observed anomalies of upper 300 m ocean heat content during MJJ in 1994 and 2006, respectively.

6. The authors should point out whether the result is an ensemble output or not. The deep learning (DL) method can be unstable to some extent. So, the result of one deep learning model is not reliable. The authors can use different number of neurons at each layer to build different DL models without changing its architecture (Ham et al 2019). Then a more complex ensemble strategy (Ham et al 2019) can be used to produce the final output.

Response: Thanks for this comment. In this study, all analyses are based on the average of 10 MLT-NET models at each lead time. We have briefly explained this in Methods (see Line 436-438) and clarified this in figure captions.

7. The authors only provided the SON skill in the main content of this study. It is widely known that the state of IOD peak season is relatively easier to predict when compared to that of other seasons (actually ENSO as well). Therefore, it is hard to evaluate the real IOD prediction improvement caused by MTL-NET. In fact, it seems that the MAM and JJA skills in supplementary information are not as excellent as the SON skill.

Response: We think that the reviewer requires to compare the prediction skills of the IOD in all seasons. In this manuscript, we evaluated the skills in MAM, JJA and SON, but not in DJF. This is because the IOD quickly demises in December and does not exist in DJF because of the reversal of East Asia-Australia monsoonal winds. The northwesterly winds along the west coast of Sumatra during DJF prohibit the local Bjerknes feedback and hence the IOD. Although the values of the IOD indices in DJF may not be zero, they basically do not represent the IOD-related physics. Note that, unlike ENSO that displays a quasi-cycle behaviour, IOD is better viewed as occasional events.

For the skill intercomparison, we mostly focused on the SON season when the IOD culminates its strongest value and hence induces strongest impacts on the regional and even global climate and environment. Hence, it is of great importance to improve the IOD forecasts in SON in order to prevent/reduce socio-economic losses. In addition, we have also compared the skills in predicting the IOD in MAM and JJA. While it becomes much harder to predict the onset and development of the IOD in MAM and JJA (as in the case of the prediction of different ENSO phases, as the reviewer pointed out), owing to the influence of stronger intra-seasonal noise and hence lower signal-to-noise ratio, our MTL-NET persistently produces better skills in predicting the IOD not only in SON but also in MAM and JJA (see Figs. 2 and S2). In summary, our MTL-

NET does show improved skills of the IOD prediction at different phases. We have briefly added the above statement (see Line 109-112).

8. The study didn't give forecasted timeseries of any target season at any lead times. The figure of forecasted timeseries is a quick and visual way to tell readers about your skills. More importantly, it also tells us the amplitude of your forecasted DMI, since the amplitude problem is a chronic disease in IOD prediction of machine learning (Ratnam et al 2020, Liu et al 2021). It will be much better if an all-season timeseries plot at different leads is shown.

Response: Following the reviewer's suggestion, we have added the observed and predicted IOD timeseries during SON, MAM and JJA at 1, 3, 6, and 12 months lead (see Fig. 2 and S2) and relevant descriptions (see Line 112-114). As was explained in the above response, we did not examine the DJF season.

9. In Figure S5, how did this study quantify the contribution of different input? The channel attention is actually a SE block (Hu et al 2017). From what I know, there exists no reliable method to separate one input layer (e.g., SST) from another (e.g., HC), since the authors stack the 2D maps (SST, HC, winds) to a cube (like a real RGB map) as input. The relations between the three input variables are highly connected in both the feature map (extracted by convolution layer) and the final attention tensor. It is now a challenge even in the computer science. So, we would like to hear more details about it.

Response: Yes, we agree with the reviewer that this is an SE-like module. In this study, this module also shows certain function to select different factors regarding their relative importance in accord with attention mechanisms (Fig. S5). The limitation of current technology in computer science impedes better solving the 'mix' problem. The

extraction of multi-modal features may help to some extent. We appreciate your suggestion, and it will be a good research direction in future studies (see Line 243-246).

Reference:

1. Ratnam, J.V., Dijkstra, H.A. & Behera, S.K. A machine learning based prediction system for the Indian Ocean Dipole. *Sci. Rep.* **10**, 284 (2020).
2. Liu, J., Tang, Y., Wu, Y., Li, T., Wang, Q. and Chen, D. Forecasting the Indian Ocean Dipole with deep learning techniques. *Geophys. Res. Lett.* **48**, e2021GL094407 (2021).
3. Cai, W., et al. Pantropical climate interactions. *Science* **363**(6430), eaav4236 (2019).
4. Kendall, A., Gal, Y., Cipolla, R. Multi-task learning using uncertainty to weigh losses for scene geometry and semantics. *Proceedings of the IEEE/CVF Conference on Computer Vision and Pattern Recognition (CVPR)* 7482-7491 (2018).
5. Liu, S., Johns, E. & Davison, A. J. End-to-end multi-task learning with attention. *Proceedings of the IEEE/CVF Conference on Computer Vision and Pattern Recognition (CVPR)* 1871-1880 (2019).
6. Venzke, S., M. Latif, Villwock, A. The coupled GCM ECHO-2. Part II: Indian Ocean response to ENSO. *J. Clim.* **13**, 1371–1383 (2000).
7. Baquero-Bernal, A., Latif, M. Stephanie, L. On dipolelike variability of sea surface temperature in the tropical Indian Ocean. *J. Clim.* **15**, 1358–1368 (2002).
8. Stuecker, M. A., et al. Revisiting ENSO/Indian Ocean Dipole phase relationships. *Geophys. Res. Lett.* **44**, 2481–2492 (2017).
9. Wang, H., et al. Covariations between the Indian Ocean dipole and ENSO: a modelling study. *Clim. Dyn.* **53**, 5743-5761 (2019).
10. Higgins, I, et al. Beta-vae: Learning basic visual concepts with a constrained variational framework. *Proc. ICLR* (2017).
11. Lin, T. Y., et al. Focal loss for dense object detection. *Proceedings of the IEEE international conference on computer vision* 2980-2988 (2017).

12. Zhu, J. Y., et al. Unpaired image-to-image translation using cycle-consistent adversarial networks. *Proceedings of the IEEE international conference on computer vision* 2223-2232 (2017).
13. Ravuri, S., et al. Skilful precipitation nowcasting using deep generative models of radar. *Nature* **597** (7878), 672-67 (2021).
14. Pan, B., et al. Learning to correct climate projection biases. *J. Adv. Model. Earth Syst.* **13** (10), e2021MS002509 (2021).
15. Ding, R., et al. Multi-year El Niño events tied to the North Pacific Oscillation. *Nat. Commun.* **13**, 3871 (2022). <https://doi.org/10.1038/s41467-022-31516-9>.
16. Bingel, J., Søgaard, A. Identifying beneficial task relations for multi-task learning in deep neural networks. Preprint at <https://arxiv.org/abs/1702.08303> (2017).
17. Sermanet, P., et al. Overfeat: Integrated recognition, localization and detection using convolutional networks. *International Conference on Learning Representations (ICLR)* (2014).
18. Kokkinos, I. UberNet: Training a 'Universal' Convolutional Neural Network for Low-, Mid-, and High-Level Vision using Diverse Datasets and Limited Memory. *30th IEEE/CVF Conference on Computer Vision and Pattern Recognition (CVPR)* (2017).
19. Eigen, D., Fergus, R. Predicting Depth, Surface Normals and Semantic Labels with a Common Multi-Scale Convolutional Architecture. *2015 IEEE International Conference on Computer Vision (ICCV)* (2014).
20. Horii, T., Hase, H., Ueki, I., Masumoto, Y. Oceanic precondition and evolution of the 2006 Indian Ocean dipole. *Geophys. Res. Lett.* **35**, (2008).
21. Luo, J.-J., Behera, S., Masumoto, Y., Sakuma, H., Yamagata, T. Successful prediction of the consecutive IOD in 2006 and 2007. *Geophys. Res. Lett.* **35**, L14S02 (2008).
22. Behera, S. K., J. J. Luo, S. Masson, S. A. Rao, H. Sakuma, and T. Yamagata, 2006: A CGCM study on the interaction between IOD and ENSO. *J. Clim.* **19**, 1688–1705, doi:10.1175/JCLI3797.1.
23. Yang, Y., et al. Seasonality and predictability of the Indian Ocean Dipole mode: ENSO forcing and internal variability. *J. Clim.* **28**, 8021–8036, (2006).
24. Behera, S. K., et al. A CGCM study on the interaction between IOD and ENSO. *J. Clim.* **19**, 1688-1705 (2006).

REVIEWERS' COMMENTS

Reviewer #1 (Remarks to the Author):

The authors have addressed my early comments/concerns in their Response to Reviewer and the revised manuscript. I therefore recommend acceptance of this manuscript for publication in Nature Communications.

Reviewer #3 (Remarks to the Author):

Review of "Multi-task Machine Learning Improves Multi-Seasonal Prediction of the Indian Ocean Dipole" by Ling et al.

Authors have addressed the concerns raised by the reviewers. The manuscript can be accepted with minor changes.

Minor Comments:

In response Q3 of Reviewer 2:

Lines 185-189: The Matsuno-Gill response is a response to anomalous heating associated with SST (not SST anomaly itself). The anomalous response to heating is such that it causes anomalous cyclonic circulation to the north-west/west of the heat source. To attribute the basin wide cyclonic circulation (Lines 185-189) to Matsuno-Gill response authors need to be sure the north Indian Ocean had a region of high precipitation (heat source) along with warm SST anomalies. In Fig S7c the cyclonic circulation is over the source and not to the west/northwest of the source.

Response to Reviewers

We would like to thank the reviewers for their constructive comments and suggestions. We have carefully addressed the reviewers' concerns in the revised manuscript. The responses are listed below in blue font.

Response to Reviewer#3

Minor comments:

In response Q3 of Reviewer 2:

Lines 185-189: The Matsuno-Gill response is a response to anomalous heating associated with SST (not SST anomaly itself). The anomalous response to heating is such that it causes anomalous cyclonic circulation to the north-west/west of the heat source. To attribute the basin wide cyclonic circulation (Lines 185-189) to Matsuno-Gill response authors need to be sure the north Indian Ocean had a region of high precipitation (heat source) along with warm SST anomalies. In Fig S7c the cyclonic circulation is over the source and not to the west/northwest of the source.

Response: Thank you for your suggestion. We have checked the precipitation anomalies and found that there are indeed positive precipitation anomalies associated with the warm SST anomalies in the tropical and North Indian Ocean (see Fig. R1). The basin-wide warm SST anomalies in the tropical Indian Ocean (not just the North Indian Ocean as was wrongly mentioned in the previous version of the manuscript) induce the large-scale cyclonic circulation with strong northerly winds in the Arabian Sea. We have rephrased this statement to clarify this point (see Line 190-193).

Figure R1. The observed precipitation anomaly in May-Jun-Jul 1998.